# Synthesis of MCMC and Belief Propagation

**Sungsoo Ahn**[*]      **Michael Chertkov**[†]      **Jinwoo Shin**[*]
[*]School of Electrical Engineering,
Korea Advanced Institute of Science and Technology, Daejeon, Korea
[†1] Theoretical Division, T-4 & Center for Nonlinear Studies,
Los Alamos National Laboratory, Los Alamos, NM 87545, USA,
[†2]Skolkovo Institute of Science and Technology, 143026 Moscow, Russia
*{sungsoo.ahn, jinwoos}@kaist.ac.kr    †chertkov@lanl.gov

## Abstract

Markov Chain Monte Carlo (MCMC) and Belief Propagation (BP) are the most
popular algorithms for computational inference in Graphical Models (GM). In
principle, MCMC is an exact probabilistic method which, however, often suffers
from exponentially slow mixing. In contrast, BP is a deterministic method, which is
typically fast, empirically very successful, however in general lacking control of ac-
curacy over loopy graphs. In this paper, we introduce MCMC algorithms correcting
the approximation error of BP, i.e., we provide a way to compensate for BP errors
via a consecutive BP-aware MCMC. Our framework is based on the Loop Calculus
approach which allows to express the BP error as a sum of weighted generalized
loops. Although the full series is computationally intractable, it is known that a trun-
cated series, summing up all 2-regular loops, is computable in polynomial-time for
planar pair-wise binary GMs and it also provides a highly accurate approximation
empirically. Motivated by this, we first propose a polynomial-time approximation
MCMC scheme for the truncated series of general (non-planar) pair-wise binary
models. Our main idea here is to use the Worm algorithm, known to provide fast
mixing in other (related) problems, and then design an appropriate rejection scheme
to sample 2-regular loops. Furthermore, we also design an efficient rejection-free
MCMC scheme for approximating the full series. The main novelty underlying
our design is in utilizing the concept of cycle basis, which provides an efficient
decomposition of the generalized loops. In essence, the proposed MCMC schemes
run on transformed GM built upon the non-trivial BP solution, and our experiments
show that this synthesis of BP and MCMC outperforms both direct MCMC and
bare BP schemes.

## 1   Introduction

GMs express factorization of the joint multivariate probability distributions in statistics via graph of
relations between variables. The concept of GM has been used successfully in information theory,
physics, artificial intelligence and machine learning [1, 2, 3, 4, 5, 6]. Of many inference problems
one can set with a GM, computing partition function (normalization), or equivalently marginalizing
the joint distribution, is the most general problem of interest. However, this paradigmatic inference
problem is known to be computationally intractable in general, i.e., formally it is #P-hard even to
approximate [7, 8].

To address this obstacle, extensive efforts have been made to develop practical approximation methods,
among which MCMC- [9] based and BP- [10] based algorithms are, arguably, the most popular
and practically successful ones. MCMC is exact, i.e., it converges to the correct answer, but its
convergence/mixing is, in general, exponential in the system size. On the other hand, message

passing implementations of BP typically demonstrate fast convergence, however in general lacking approximation guarantees for GM containing loops. Motivated by this complementarity of the MCMC and BP approaches, we aim here to synthesize a hybrid approach benefiting from a joint use of MCMC and BP.

At a high level, our proposed scheme uses BP as the first step and then runs MCMC to correct for the approximation error of BP. To design such an "error-correcting" MCMC, we utilize the Loop Calculus approach [11] which allows, in a nutshell, to express the BP error as a sum (i.e., series) of weights of the so-called generalized loops (sub-graphs of a special structure). There are several challenges one needs to overcome. First of all, to design an efficient Markov Chain (MC) sampler, one needs to design a scheme which allows efficient transitions between the generalized loops. Second, even if one designs such a MC which is capable of accessing all the generalized loops, it may mix slowly. Finally, weights of generalized loops can be positive or negative, while an individual MCMC can only generate non-negative contributions.

Since approximating the full loop series (LS) is intractable in general, we first explore whether we can deal with the challenges at least in the case of the truncated LS corresponding to 2-regular loops. In fact, this problem has been analyzed in the case of the planar pairwise binary GMs [12, 13] where it was shown that the 2-regular LS is computable exactly in polynomial-time through a reduction to a Pfaffian (or determinant) computation [14]. In particular, the partition function of the Ising model without external field (i.e., where only pair-wise factors present) is computable exactly via the 2-regular LS. Furthermore, the authors show that in the case of general planar pairwise binary GMs, the 2-regular LS provides a highly accurate approximation empirically. Motivated by these results, we address the same question in the general (i.e., non-planar) case of pairwise binary GMs via MCMC. For the choice of MC, we adopt the Worm algorithm [15]. We prove that with some modification including rejections, the algorithm allows to sample (with probabilities proportional to respective weights) 2-regular loops in polynomial-time. Then, we design a novel simulated annealing strategy using the sampler to estimate separately positive and negative parts of the 2-regular LS. Given any $\varepsilon > 0$, this leads to a $\varepsilon$-approximation polynomial-time scheme for the 2-regular LS under a mild assumption.

We next turn to estimating the full LS. In this part, we ignore the theoretical question of establishing the polynomial mixing time of a MC, and instead focus on designing an empirically efficient MCMC scheme. We design an MC using a cycle basis of the graph [16] to sample generalized loops directly, without rejections. It transits from one generalized loop to another by adding or deleting a random element of the cycle basis. Using the MC sampler, we design a simulated annealing strategy for estimating the full LS, which is similar to what was used earlier to estimate the 2-regular LS. Notice that even though the prime focus of this paper is on pairwise binary GMs, the proposed MCMC scheme allows straightforward generalization to general non-binary GMs.

In summary, we propose novel MCMC schemes to estimate the LS correction to the BP contribution to the partition function. Since already the bare BP provides a highly non-trivial estimation for the partition function, it is naturally expected and confirmed in our experimental results that the proposed algorithm outperforms other standard (not related to BP) MCMC schemes applied to the original GM. We believe that our approach provides a new angle for approximate inference on GM and is of broader interest to various applications involving GMs.

## 2 Preliminaries

### 2.1 Graphical models and belief propagation

Given undirected graph $G = (V, E)$ with $|V| = n, |E| = m$, a pairwise binary *Markov Random Fields* (MRF) defines the following joint probability distribution on $x = [x_v \in \{0, 1\} : v \in V]$:

$$p(x) = \frac{1}{Z} \prod_{v \in V} \psi_v(x_v) \prod_{(u,v) \in E} \psi_{u,v}(x_u, x_v), \qquad Z := \sum_{x \in \{0,1\}^n} \prod_{v \in V} \psi_v(x_v) \prod_{(u,v) \in E} \psi_{u,v}, (x_u, x_v)$$

where $\psi_v, \psi_{u,v}$ are some non-negative functions, called *compatibility* or *factor* functions, and the normalization constant $Z$ is called the *partition function*. Without loss of generality, we assume $G$ is connected. It is known that approximating the partition function is #P-hard in general [8]. Belief Propagation (BP) is a popular message-passing heuristic for approximating marginal distributions of

MRF. The BP algorithm iterates the following message updates for all $(u, v) \in E$:

$$m_{u \to v}^{t+1}(x_v) \propto \sum_{x_u \in \{0,1\}} \psi_{u,v}(x_u, x_v) \psi_u(x_u) \prod_{w \in N(u) \backslash v} m_{w \to u}^t(x_u),$$

where $N(v)$ denotes the set of neighbors of $v$. In general BP may fail to converge, however in this case one may substitute it with a somehow more involved algorithm provably convergent to its fixed point [22, 23, 24]. Estimates for the marginal probabilities are expressed via the fixed-point messages $\{m_{u \to v} : (u, v) \in E\}$ as follows: $\tau_v(x_v) \propto \psi_v(x_v) \prod_{u \in N(v)} m_{u \to v}(x_v)$ and

$$\tau_{u,v}(x_u, x_v) \propto \psi_u(x_u) \psi_v(x_v) \psi_{u,v}(x_u, x_v) \left( \prod_{w \in N(u)} m_{w \to v}(x_u) \right) \left( \prod_{w \in N(v)} m_{w \to v}(x_v) \right).$$

## 2.2 Bethe approximation and loop calculus

BP marginals also results in the following *Bethe approximation* for the partition function $Z$:

$$\log Z_{\text{Bethe}} = \sum_{v \in V} \sum_{x_v} \tau_v(x_v) \log \psi_v(x_v) + \sum_{(u,v) \in E} \sum_{x_u, x_v} \tau_{u,v}(x_u, x_v) \log \psi_{u,v}(x_u, x_v)$$
$$- \sum_{v \in V} \sum_{x_v} \tau_v(x_v) \log \tau_v(x_v) - \sum_{(u,v) \in E} \sum_{x_u, x_v} \tau_{u,v}(x_u, x_v) \log \frac{\tau_{u,v}(x_u, x_v)}{\tau_u(x_u) \tau_v(x_v)}$$

If graph $G$ is a tree, the Bethe approximation is exact, i.e., $Z_{\text{Bethe}} = Z$. However, in general, i.e. for the graph with cycles, BP algorithm provides often rather accurate but still an approximation.

*Loop Series* (LS) [11] expresses, $Z/Z_{\text{Bethe}}$, as the following sum/series:

$$\frac{Z}{Z_{\text{Bethe}}} = Z_{\text{Loop}} := \sum_{F \in \mathcal{L}} w(F), \quad w(\emptyset) = 1,$$

$$w(F) := \prod_{(u,v) \in E_F} \left( \frac{\tau_{u,v}(1,1)}{\tau_u(1)\tau_v(1)} - 1 \right) \prod_{v \in V_F} \left( \tau_v(1) + (-1)^{d_F(v)} \left( \frac{\tau_v(1)}{1 - \tau_v(1)} \right)^{d_F(v)-1} \tau_v(1) \right)$$

where each term/weight is associated with the so-called *generalized loop* $F$ and $\mathcal{L}$ denotes the set of all generalized loops in graph $G$ (including the empty subgraph $\emptyset$). Here, a subgraph $F$ of $G$ is called generalized loop if all vertices $v \in F$ have degree $d_F(v)$ (in the subgraph) no smaller than 2.

Since the number of generalized loops is exponentially large, computing $Z_{\text{Loop}}$ is intractable in general. However, the following truncated sum of $Z_{\text{Loop}}$, called *2-regular loop series*, is known to be computable in polynomial-time if $G$ is planar [12]:[1]

$$Z_{\text{2-Loop}} := \sum_{F \in \mathcal{L}_{\text{2-Loop}}} w(F),$$

where $\mathcal{L}_{\text{2-Loop}}$ denotes the set of all *2-regular generalized loops*, i.e., $F \in \mathcal{L}_{\text{2-Loop}}$ if $d_F(v) = 2$ for every vertex $v$ of $F$. One can check that $Z_{\text{Loop}} = Z_{\text{2-Loop}}$ for the Ising model without the external fields. Furthermore, as stated in [12, 13] for the general case, $Z_{\text{2-Loop}}$ provides a good empirical estimation for $Z_{\text{Loop}}$.

## 3 Estimating 2-regular loop series via MCMC

In this section, we aim to describe how the 2-regular loop series $Z_{\text{2-Loop}}$ can be estimated in polynomial-time. To this end, we first assume that the maximum degree $\Delta$ of the graph $G$ is at most 3. This degree constrained assumption is not really restrictive since any pairwise binary model can be easily expressed as an equivalent one with $\Delta \leq 3$, e.g., see the supplementary material.

The rest of this section consists of two parts. We first propose an algorithm generating a 2-regular loop sample with the probability proportional to the absolute value of its weight, i.e.,

$$\pi_{\text{2-Loop}}(F) := \frac{|w(F)|}{Z^{\dagger}_{\text{2-Loop}}}, \qquad \text{where } Z^{\dagger}_{\text{2-Loop}} = \sum_{F \in \mathcal{L}_{\text{2-Loop}}} |w(F)|.$$

Note that this 2-regular loop contribution allows the following factorization: for any $F \in \mathcal{L}_{\text{2-Loop}}$,

$$|w(F)| = \prod_{e \in F} w(e), \qquad \text{where} \quad w(e) := \left| \frac{\tau_{u,v}(1,1) - \tau_u(1)\tau_v(1)}{\sqrt{\tau_u(1)\tau_v(1)(1 - \tau_u(1))(1 - \tau_v(1))}} \right|. \qquad (1)$$

In the second part we use the sampler constructed in the first part to design a simulated annealing scheme to estimate $Z_{\text{2-Loop}}$.

### 3.1  Sampling 2-regular loops

We suggest to sample the 2-regular loops distributed according to $\pi_{\text{2-Loop}}$ through a version of the Worm algorithm proposed by Prokofiev and Svistunov [15]. It can be viewed as a MC exploring the set, $\mathcal{L}_{\text{2-Loop}} \bigcup \mathcal{L}_{\text{2-Odd}}$, where $\mathcal{L}_{\text{2-Odd}}$ is the set of all subgraphs of $G$ with exactly two odd-degree vertices. Given current state $F \in \mathcal{L}_{\text{2-Loop}} \bigcup \mathcal{L}_{\text{2-Odd}}$, it chooses the next state $F'$ as follows:

1. If $F \in \mathcal{L}_{\text{2-Odd}}$, pick a random vertex $v$ (uniformly) from $V$. Otherwise, pick a random odd-degree vertex $v$ (uniformly) from $F$.

2. Choose a random neighbor $u$ of $v$ (uniformly) within $G$, and set $F' \leftarrow F$ initially.

3. Update $F' \leftarrow F \oplus \{u, v\}$ with the probability

$$\begin{cases} \min\left( \frac{1}{n} \frac{|w(F \oplus \{u,v\})|}{|w(F)|}, 1 \right) & \text{if } F \in \mathcal{L}_{\text{2-Loop}} \\ \min\left( \frac{n}{4} \frac{|w(F \oplus \{u,v\})|}{|w(F)|}, 1 \right) & \text{else if } F \oplus \{u, v\} \in \mathcal{L}_{\text{2-Loop}} \\ \min\left( \frac{d(v)}{2d(u)} \frac{|w(F \oplus \{u,v\})|}{|w(F)|}, 1 \right) & \text{else if } F, F \oplus \{u, v\} \in \mathcal{L}_{\text{2-Odd}} \end{cases}$$

Here, $\oplus$ denotes the symmetric difference and for $F \in \mathcal{L}_{\text{2-Odd}}$, its weight is defined according to $w(F) = \prod_{e \in F} w(e)$. In essence, the Worm algorithm consists in either deleting or adding an edge to the current subgraph $F$. From the Worm algorithm, we transition to the following algorithm which samples 2-regular loops with probability $\pi_{\text{2-Loop}}$ simply by adding rejection of $F$ if $F \in \mathcal{L}_{\text{2-Odd}}$.

---

**Algorithm 1** Sampling 2-regular loops

1: **Input:** Number of trials $N$; number of iterations $T$ of the Worm algorithm
2: **Output:** 2-regular loop $F$.
3: **for** $i = 1 \rightarrow N$ **do**
4:     Set $F \leftarrow \emptyset$ and update it $T$ times by running the Worm algorithm
5:     **if** $F$ is a 2-regular loop **then**
6:         BREAK and output $F$.
7:     **end if**
8: **end for**
9: Output $F = \emptyset$.

---

The following theorem states that Algorithm 1 can generate a desired random sample in polynomial-time.

**Theorem 1.** *Given $\delta > 0$, choose inputs of Algorithm 1 as*

$$N \geq 1.2\, n \log(3\delta^{-1}), \qquad and \qquad T \geq (m - n + 1)\log 2 + 4\Delta mn^4 \log(3n\delta^{-1}).$$

*Then, it follows that*

$$\frac{1}{2} \sum_{F \in \mathcal{L}_{\text{2-Loop}}} \left| P\Big[\textit{Algorithm 1 outputs } F\Big] - \pi_{\text{2-Loop}}(F) \right| \leq \delta.$$

*namely, the total variation distance between $\pi_{\text{2-Loop}}$ and the output distribution of Algorithm 1 is at most $\delta$.*

The proof of the above theorem is presented in the supplementary material due to the space constraint. In the proof, we first show that MC induced by the Worm algorithm mixes in polynomial time, and then prove that acceptance of a 2-regular loop, i.e., line 6 of Algorithm 1, occurs with high probability. Notice that the uniform-weight version of the former proof, i.e., fast mixing, was recently proven in [18]. For completeness of the material exposition, we present the general case proof of interest for us. The latter proof, i.e., high acceptance, requires to bound $|\mathcal{L}_{\text{2-Loop}}|$ and $|\mathcal{L}_{\text{2-Odd}}|$ to show that the probability of sampling 2-regular loops under the Worm algorithm is $1/\text{poly}(n)$ for some polynomial function $\text{poly}(n)$.

## 3.2 Simulated annealing for approximating 2-regular loop series

Here we utilize Theorem 1 to describe an algorithm approximating the 2-regular LS $Z_{\text{2-Loop}}$ in polynomial time. To achieve this goal, we rely on the simulated annealing strategy [19] which requires to decide a monotone cooling schedule $\beta_0, \beta_1, \ldots, \beta_{\ell-1}, \beta_\ell$, where $\beta_\ell$ corresponds to the target counting problem and $\beta_0$ does to its relaxed easy version. Thus, designing an appropriate cooling strategy is the first challenge to address. We will also describe how to deal with the issue that $Z_{\text{2-Loop}}$ is a sum of positive and negative terms, while most simulated annealing strategies in the literature mainly studied on sums of non-negative terms. This second challenge is related to the so-called 'fermion sign problem' common in statistical mechanics of quantum systems [25]. Before we describe the proposed algorithm in details, let us provide its intuitive sketch.

The proposed algorithm consists of two parts: a) estimating $Z^{\dagger}_{\text{2-Loop}}$ via a simulated annealing strategy and b) estimating $Z_{\text{2-Loop}}/Z^{\dagger}_{\text{2-Loop}}$ via counting samples corresponding to negative terms in the 2-regular loop series. First consider the following $\beta$-parametrized, auxiliary distribution over 2-regular loops:

$$\pi_{\text{2-Loop}}(F : \beta) = \frac{1}{Z^{\dagger}_{\text{2-Loop}}(\beta)} |w(F)|^{\beta}, \qquad \text{for } 0 \le \beta \le 1. \qquad (2)$$

Note that one can generate samples approximately with probability (2) in polynomial-time using Algorithm 1 by setting $w \leftarrow w^{\beta}$. Indeed, it follows that for $\beta' > \beta$,

$$\frac{Z^{\dagger}_{\text{2-Loop}}(\beta')}{Z^{\dagger}_{\text{2-Loop}}(\beta)} = \sum_{F \in \mathcal{L}_{\text{2-Loop}}} |w(F)|^{\beta'-\beta} \frac{|w(F)|^{\beta}}{Z^{\dagger}_{\text{2-Loop}}(\beta)} = \mathbb{E}_{\pi_{\text{2-Loop}}(\beta)} \left[ |w(F)|^{\beta'-\beta} \right],$$

where the expectation can be estimated using $O(1)$ samples if it is $\Theta(1)$, i.e., $\beta'$ is sufficiently close to $\beta$. Then, for any increasing sequence $\beta_0 = 0, \beta_1, \ldots, \beta_{n-1}, \beta_n = 1$, we derive

$$Z^{\dagger}_{\text{2-Loop}} = \frac{Z^{\dagger}_{\text{2-Loop}}(\beta_n)}{Z^{\dagger}_{\text{2-Loop}}(\beta_{n-1})} \cdot \frac{Z^{\dagger}_{\text{2-Loop}}(\beta_{n-1})}{Z^{\dagger}_{\text{2-Loop}}(\beta_{n-2})} \cdots \frac{Z^{\dagger}_{\text{2-Loop}}(\beta_2)}{Z^{\dagger}_{\text{2-Loop}}(\beta_1)} \frac{Z^{\dagger}_{\text{2-Loop}}(\beta_1)}{Z^{\dagger}_{\text{2-Loop}}(\beta_0)} Z^{\dagger}_{\text{2-Loop}}(0),$$

where it is know that $Z^{\dagger}_{\text{2-Loop}}(0)$, i.e., the total number of 2-regular loops, is exactly $2^{m-n+1}$ [16]. This allows us to estimate $Z^{\dagger}_{\text{2-Loop}}$ simply by estimating $\mathbb{E}_{\pi_{\text{2-Loop}}(\beta_i)} \left[ |w(F)|^{\beta_{i+1}-\beta_i} \right]$ for all $i$.

Our next step is to estimate the ratio $Z_{\text{2-Loop}}/Z^{\dagger}_{\text{2-Loop}}$. Let $\mathcal{L}^{-}_{\text{2-Loop}}$ denote the set of negative 2-regular loops, i.e.,

$$\mathcal{L}^{-}_{\text{2-Loop}} := \{F : F \in \mathcal{L}_{\text{2-Loop}}, w(F) < 0\}.$$

Then, the 2-regular loop series can be expressed as

$$Z_{\text{2-Loop}} = \left( 1 - 2 \frac{\sum_{F \in \mathcal{L}^{-}_{\text{2-Loop}}} |w(F)|}{Z^{\dagger}_{\text{2-Loop}}} \right) Z^{\dagger}_{\text{2-Loop}} = \left( 1 - 2 P_{\pi_{\text{2-Loop}}} \left[ w(F) < 0 \right] \right) Z^{\dagger}_{\text{2-Loop}},$$

where we estimate $P_{\pi_{\text{2-Loop}}} \left[ w(F) < 0 \right]$ again using samples generated by Algorithm 1.

We provide the formal description of the proposed algorithm and its error bound as follows.

---

**Algorithm 2** Approximation for $Z_{\text{2-Loop}}$

---

1: **Input:** Increasing sequence $\beta_0 = 0 < \beta_1 < \cdots < \beta_{n-1} < \beta_n = 1$; number of samples $s_1, s_2$; number of trials $N_1$; number of iterations $T_1$ for Algorithm 1.
2: **for** $i = 0 \to n - 1$ **do**
3:     Generate 2-regular loops $F_1, \dots, F_{s_1}$ for $\pi_{\text{2-Loop}}(\beta_i)$ using Algorithm 1 with input $N_1$ and $T_1$, and set

$$H_i \leftarrow \frac{1}{s_1} \sum_j w(F_j)^{\beta_{i+1} - \beta_i}.$$

4: **end for**
5: Generate 2-regular loops $F_1, \dots, F_{s_2}$ for $\pi_{\text{2-Loop}}$ using Algorithm 1 with input $N_2$ and $T_2$, and set

$$\kappa \leftarrow \frac{|\{F_j : w(F_j) < 0\}|}{s_2}.$$

6: **Output:** $\widehat{Z}_{\text{2-Loop}} \leftarrow (1 - 2\kappa) 2^{m-n+1} \prod_i H_i.$

---

**Theorem 2.** *Given $\varepsilon, \nu > 0$, choose inputs of Algorithm 2 as $\beta_i = i/n$ for $i = 1, 2, \dots, n-1$,*

$$s_1 \geq 18144 n^2 \varepsilon^{-2} w_{\min}^{-1} \lceil \log(6n\nu^{-1}) \rceil, \qquad\qquad N_1 \geq 1.2n \log(144 n \varepsilon^{-1} w_{\min}^{-1}),$$

$$T_1 \geq (m - n + 1) \log 2 + 4\Delta m n^4 \log(48 n \varepsilon^{-1} w_{\min}^{-1}),$$

$$s_2 \geq 18144 \zeta (1 - 2\zeta)^{-2} \varepsilon^{-2} \lceil \log(3\nu^{-1}) \rceil, \qquad\qquad N_2 \geq 1.2n \log(144 \varepsilon^{-1} (1 - 2\zeta)^{-1}),$$

$$T_2 \geq (m - n + 1) \log 2 + 4\Delta m n^4 \log(48 \varepsilon^{-1} (1 - 2\zeta)^{-1})$$

*where $w_{\min} = \min_{e \in E} w(e)$ and $\zeta = P_{\pi_{\text{2-Loop}}}[w(F) < 0]$. Then, the following statement holds*

$$P\left[ \frac{|\widehat{Z}_{\text{2-Loop}} - Z_{\text{2-Loop}}|}{Z_{\text{2-Loop}}} \leq \varepsilon \right] \leq 1 - \nu,$$

*which means Algorithm 2 estimates $Z_{\text{2-Loop}}$ within approximation ratio $1 \pm \varepsilon$ with high probability.*

The proof of the above theorem is presented in the supplementary material due to the space constraint. We note that all constants entering in Theorem 2 were not optimized. Theorem 2 implies that complexity of Algorithm 2 is polynomial with respect to $n, 1/\varepsilon, 1/\nu$ under assumption that $w_{\min}^{-1}$ and $1 - 2P_{\pi_{\text{2-Loop}}}[w(F) < 0]$ are polynomially small. Both $w_{\min}^{-1}$ and $1 - 2P_{\pi_{\text{2-Loop}}}[w(F) < 0]$ depend on the choice of BP fixed point, however it is unlikely (unless a degeneracy) that these characteristics become large. In particular, $P_{\pi_{\text{2-Loop}}}[w(F) < 0] = 0$ in the case of attractive models [20].

## 4 Estimating full loop series via MCMC

In this section, we aim for estimating the full loop series $Z_{\text{Loop}}$. To this end, we design a novel MC sampler for generalized loops, which adds (or removes) a cycle basis or a path to (or from) the current generalized loop. Therefore, we naturally start this section introducing necessary backgrounds on *cycle basis*. Then, we turn to describe the design of MC sampler for generalized loops. Finally, we describe a simulated annealing scheme similar to the one described in the preceding section. We also report its experimental performance comparing with other methods.

### 4.1 Sampling generalized loops with cycle basis

The cycle basis $\mathcal{C}$ of the graph $G$ is a minimal set of cycles which allows to represent every Eulerian subgraph of $G$ (i.e., subgraphs containing no odd-degree vertex) as a symmetric difference of cycles in the set [16]. Let us characterize the combinatorial structure of the generalized loop using the cycle basis. To this end, consider a set of paths between any pair of vertices:

$$\mathcal{P} = \{P_{u,v} : u \neq v, u, v \in V, P_{u,v} \text{ is a path from } u \text{ to } v\},$$

i.e., $|\mathcal{P}| = \binom{n}{2}$. Then the following theorem allows to decompose any generalized loop with respect to any selected $\mathcal{C}$ and $\mathcal{P}$.

**Theorem 3.** *Consider any cycle basis $\mathcal{C}$ and path set $\mathcal{P}$. Then, for any generalized loop $F$, there exists a decomposition, $\mathcal{B} \subset \mathcal{C} \cup \mathcal{P}$, such that $F$ can be expressed as a symmetric difference of the elements of $\mathcal{B}$, i.e., $F = B_1 \oplus B_2 \oplus \cdots B_{k-1} \oplus B_k$ for some $B_i \in \mathcal{B}$.*

The proof of the above theorem is given in the supplementary material due to the space constraint. Now given any choice of $\mathcal{C}, \mathcal{P}$, consider the following transition from $F \in \mathcal{L}$, to the next state $F'$:

1. Choose, uniformly at random, an element $B \in \mathcal{C} \cup \mathcal{P}$, and set $F' \leftarrow F$ initially.

2. If $F \oplus B \in \mathcal{L}$, update $F' \leftarrow \begin{cases} F \oplus B & \text{with probability } \min\left\{1, \frac{|w(F\oplus B)|}{|w(F)|}\right\} \\ F & \text{otherwise} \end{cases}$.

Due to Theorem 3, it is easy to check that the proposed MC is irreducible and aperiodic, i.e., ergodic, and the distribution of its $t$-th state converges to the following stationary distribution as $t \to \infty$:

$$\pi_{\text{Loop}}(F) = \frac{|w(F)|}{Z^\dagger_{\text{Loop}}}, \qquad \text{where } Z^\dagger_{\text{Loop}} = \sum_{F \in \mathcal{L}_{\text{Loop}}} |w(F)|.$$

One also has a freedom in choosing $\mathcal{C}, \mathcal{P}$. To accelerate mixing of MC, we suggest to choose the minimum weighted cycle basis $\mathcal{C}$ and the shortest paths $\mathcal{P}$ with respect to the edge weights $\{\log w(e)\}$ defined in (1), which are computable using the algorithm in [16] and the Bellman-Ford algorithm [21], respectively. This encourages transitions between generalized loops with similar weights.

### 4.2 Simulated annealing for approximating full loop series

---

**Algorithm 3** Approximation for $Z_{\text{Loop}}$

---

1: **Input:** Decreasing sequence $\beta_0 > \beta_1 > \cdots > \beta_{\ell-1} > \beta_\ell = 1$; number of samples $s_0, s_1, s_2$; number of iterations $T_0, T_1, T_2$ for the MC described in Section 4.1
2: Generate generalized loops $F_1, \cdots, F_{s_0}$ by running $T_0$ iterations of the MC described in Section 4.1 for $\pi_{\text{Loop}}(\beta_0)$, and set
$$U \leftarrow \frac{s_0}{s^*}|w(F^*)|^{\beta_0},$$
   where $F^* = \arg\max_{F \in \{F_1, \cdots, F_{s_0}\}} |w(F)|$ and $s^*$ is the number of $F^*$ sampled.
3: **for** $i = 0 \to \ell - 1$ **do**
4:     Generate generalized loops $F_1, \cdots, F_{s_1}$ by running $T_1$ iterations of the MC described in Section 4.1 for $\pi_{\text{Loop}}(\beta_i)$, and set $H_i \leftarrow \frac{1}{s_1}\sum_j |w(F_j)|^{\beta_{i+1}-\beta_i}$.
5: **end for**
6: Generate generalized loops $F_1, \cdots F_{s_2}$ by running $T_2$ iterations of the MC described in Section 4.1 for $\pi_{\text{Loop}}$, and set
$$\kappa \leftarrow \frac{|\{F_j : w(F_j) < 0\}|}{s_2}.$$
7: **Output:** $\widehat{Z}_{\text{Loop}} \leftarrow (1 - 2\kappa)\prod_i H_i U$.

---

Now we are ready to describe a simulated annealing scheme for estimating $Z_{\text{Loop}}$. It is similar, in principle, with that in Section 3.2. First, we again introduce the following $\beta$-parametrized, auxiliary probability distribution $\pi_{\text{Loop}}(F : \beta) = |w(F)|^\beta / Z^\dagger_{\text{Loop}}(\beta)$. For any decreasing sequence of annealing parameters, $\beta_0, \beta_1, \cdots, \beta_{\ell-1}, \beta_\ell = 1$, we derive

$$Z^\dagger_{\text{Loop}} = \frac{Z^\dagger_{\text{Loop}}(\beta_\ell)}{Z^\dagger_{\text{Loop}}(\beta_{\ell-1})} \cdot \frac{Z^\dagger_{\text{Loop}}(\beta_{\ell-1})}{Z^\dagger_{\text{Loop}}(\beta_{\ell-2})} \cdots \frac{Z^\dagger_{\text{Loop}}(\beta_2)}{Z^\dagger_{\text{Loop}}(\beta_1)} \cdot \frac{Z^\dagger_{\text{Loop}}(\beta_1)}{Z^\dagger_{\text{Loop}}(\beta_0)} Z^\dagger_{\text{Loop}}(\beta_0).$$

Following similar procedures in Section 3.2, one can estimate $Z^\dagger_{\text{Loop}}(\beta')/Z^\dagger_{\text{Loop}}(\beta) = \mathbb{E}_{\pi_{\text{Loop}}(\beta)}[|w(F)|^{\beta'-\beta}]$ using the sampler described in Section 4.1. Moreover, $Z^\dagger_{\text{Loop}}(\beta_0) = |w(F^*)|/P_{\pi_{\text{Loop}}(\beta_0)}(F^*)$ is estimated by sampling generalized loop $F^*$ with the highest probability $P_{\pi_{\text{Loop}}(\beta_0)}(F^*)$. For large enough $\beta_0$, the approximation error becomes relatively small since $P_{\pi_{\text{Loop}}(\beta_0)}(F^*) \propto |w(F^*)|^{\beta_0}$ dominates over the distribution. In combination, this provides a desired approximation for $Z_{\text{Loop}}$. The result is stated formally in Algorithm 3.

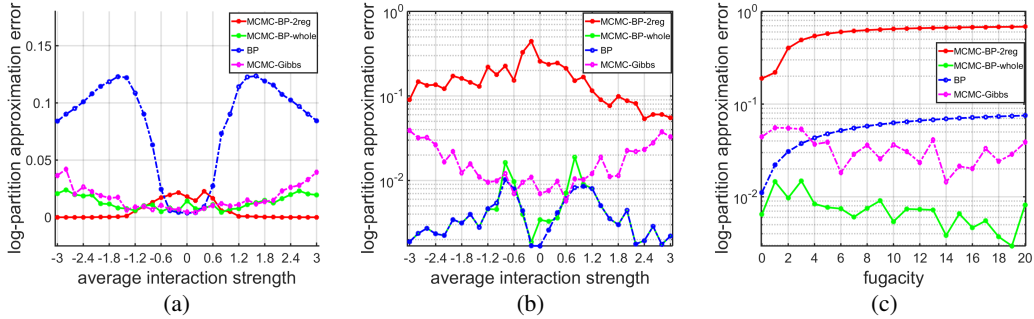

Figure 1: Plots of the log-partition function approximation error with respect to (average) interaction strength: (a) Ising model with no external field, (b) Ising model with external fields and (c) Hard-core model. Each point is averaged over 20 (random) models.

## 4.3   Experimental results

In this section, we report experimental results for computing partition function of the Ising model and the hard-core model. We compare Algorithm 2 in Section 3 (coined MCMC-BP-2reg) and Algorithm 3 in Section 4.2 (coined MCMC-BP-whole), with the bare Bethe approximation (coined BP) and the popular Gibbs-sampler (coined MCMC-Gibbs). To make the comparison fair, we use the same annealing scheme for all MCMC schemes, thus making their running times comparable. More specifically, we generate each sample after running $T_1 = 1,000$ iterations of an MC and take $s_1 = 100$ samples to compute each estimation (e.g., $H_i$) at intermediate steps. For performance measure, we use the log-partition function approximation error defined as $|\log Z - \log Z_{\text{approx}}|/|\log Z|$, where $Z_{\text{approx}}$ is the output of the respective algorithm. We conducted 3 experiments on the $4 \times 4$ grid graph. In our first experimental setting, we consider the Ising model with varying interaction strength and no external (magnetic) field. To prepare the model of interest, we start from the Ising model with uniform (ferromagnetic/attractive and anti-ferromagnetic/repulsive) interaction strength and then add 'glassy' variability in the interaction strength modeled via i.i.d Gaussian random variables with mean 0 and variance $0.5^2$, i.e. $\mathcal{N}(0, 0.5^2)$. In other words, given average interaction strength 0.3, each interaction strength in the model is independently chosen as $\mathcal{N}(0.3, 0.5^2)$. The second experiment was conducted by adding $\mathcal{N}(0, 0.5^2)$ corrections to the external fields under the same condition as in the first experiment. In this case we observe that BP often fails to converge, and use the Concave Convex Procedure (CCCP) [23] for finding BP fixed points. Finally, we experiment with the hard-core model on the $4 \times 4$ grid graph with varying a positive parameter $\lambda > 0$, called 'fugacity' [26]. As seen clearly in Figure 1, BP and MCMC-Gibbs are outperformed by MCMC-BP-2reg or MCMC-BP-whole at most tested regimes in the first experiment with no external field, where in this case, the 2-regular loop series (LS) is equal to the full one. Even in the regimes where MCMC-Gibbs outperforms BP, our schemes correct the error of BP and performs at least as good as MCMC-Gibbs. In the experiments, we observe that advantage of our schemes over BP is more pronounced when the error of BP is large. A theoretical reasoning behind this observation is as follows. If the performance of BP is good, i.e. the loop series (LS) is close to 1, the contribution of empty generalized loop, i.e., $w(\emptyset)$, in LS is significant, and it becomes harder to sample other generalized loops accurately.

## 5   Conclusion

In this paper, we propose new MCMC schemes for approximate inference in GMs. The main novelty of our approach is in designing BP-aware MCs utilizing the non-trivial BP solutions. In experiments, our BP based MCMC scheme also outperforms other alternatives. We anticipate that this new technique will be of interest to many applications where GMs are used for statistical reasoning.

**Acknowledgement**

This work was supported by the National Research Council of Science & Technology (NST) grant by the Korea government (MSIP) (No. CRC-15-05-ETRI), and funding from the U.S. Department of Energy's Office of Electricity as part of the DOE Grid Modernization Initiative.

## Footnotes

[1] Note that the number of 2-regular loops is exponentially large in general.

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
