[Supplementary Material]

## A Transformation to an equivalent binary pairwise model with maximum degree at most 3

Figure 2: Demonstration of building an equivalent model with maximum degree $\Delta \leq 3$ via 'expanding' vertices (in grey). In the new model, one can introduce edge factor $\psi_{u,v}$ between the duplicated vertices $u, v$ (in bold) such that $\psi_{u,v}(x_u, x_v) = 1$ if $x_u = x_v$ and $\psi_{u,v}(x_u, x_v) = 0$ otherwise.

## B Proof of Theorem 1

First, note that the MC induced by the worm algorithm converges to the following stationary distribution

$$\pi_{\text{WA}}(F) \propto \Psi(F) \prod_{e \in F} w(e),$$

where

$$\Psi(F) = \begin{cases} n, & \forall F \in \mathcal{L}_{\text{2-Loop}}, \\ 2, & \forall F \in \mathcal{L}_{\text{2-Odd}}. \end{cases}$$

We first prove its polynomial mixing, i.e. it produces a sample from a distribution with the desired total variation distance from $\pi_{\text{WA}}$ in a polynomial number of iterations.

**Lemma 1.** *Given any $\delta > 0$ and any $F_0 \in \mathcal{L}_{\text{2-Loop}} \cup \mathcal{L}_{\text{2-Loop}}$, choose*

$$T_{mix} \geq w(F_0)^{-1} + (m - n + 1)\log 2 + 12\Delta mn^4 \log \delta^{-1},$$

*and let $\pi_{WA}^t(\cdot)$ denote the resulting distribution of after updating $t$ times by the worm algorithm with initial state $F_0$. Then, it follows that*

$$\frac{1}{2} \sum_{F \in \mathcal{L}_{\text{2-Loop}} \cup \mathcal{L}_{\text{2-Loop}}} \left| \pi_{WA}^{T_{mix}}(F) - \pi_{WA}(F) \right| \leq \delta,$$

*namely, the mixing time of the MC is bounded above by $T_{mix}$.*

The proof of the above lemma is given in Section B.1. Collevecchio et al. [18] recently proved that the worm algorithm mixes in polynomial time when the weights are uniform, i.e., equal. We extend the result to our case of non-uniform weights. The proof is based on the method of *canonical path*, which views the state space as a graph and constructs a path between every pair of states having certain amount of flow defined by $\pi_{\text{WA}}$. From Lemma 1 with parameters

$$N \leq 1.2n \log(3\delta^{-1}), \quad T \leq (m - n + 1)\log 2 + 4\Delta mn^4 \log(3n\delta^{-1}), \quad \text{and} \quad F_0 \leftarrow \emptyset,$$

we obtain that the total variation distance between $\pi_{\text{WA}}$ and the distribution of updated states in line 4 of Algorithm 1 is at most $\frac{\delta}{3n}$. Next, we prove that the probability of acceptance in line 6 of Algorithm 1 is sufficiently large.

**Lemma 2.** *The probability of sampling a 2-regular loop from distribution $\pi_{WA}$ is bounded below by $n^{-1}$, i.e. $\pi_{WA}(\mathcal{L}_{\text{2-Loop}}) \geq \frac{1}{n}$.*

The proof of the above lemma is given in Section B.2. The proof relies on the fact that the size of $\mathcal{L}_{\text{2-Loop}}$ is bounded by a polynomial of the size of $\mathcal{L}_{\text{2-Odd}}$.

Now we are ready to complete the proof of Theorem 1. Let $\widehat{\pi}_{\text{2-Loop}}$ denote the distribution of 2-regular loops from line 6 of Algorithm 1 under parameters as in Theorem 1. We say Algorithm 1 fails if it

outputs $F = \emptyset$ from line 9. Choose a set of 2-regular loops $\widehat{\mathcal{L}}_{\text{2-Loop}} := \{F \in \mathcal{L}_{\text{2-Loop}} : \widehat{\pi}_{\text{2-Loop}}(F) > \pi_{\text{2-Loop}}(F)\}$. Then the total variation distance between $\pi_{\text{2-Loop}}$ and $\widehat{\pi}_{\text{2-Loop}}$ can be expressed as:

$$\frac{1}{2} \sum_{F \in \mathcal{L}_{\text{2-Loop}}} |\widehat{\pi}_{\text{2-Loop}}(F) - \pi_{\text{2-Loop}}(F)| = \widehat{\pi}_{\text{2-Loop}}(\widehat{\mathcal{L}}_{\text{2-Loop}}) - \pi_{\text{2-Loop}}(\widehat{\mathcal{L}}_{\text{2-Loop}}).$$

By applying Lemma 1 and Lemma 2, we obtain the following under parameters as in Theorem 1:

$$\widehat{\pi}_{\text{2-Loop}}(\widehat{\mathcal{L}}_{\text{2-Loop}}) - \pi_{\text{2-Loop}}(\widehat{\mathcal{L}}_{\text{2-Loop}})$$

$$\overset{(a)}{\geq} \frac{\widehat{\pi}_{\text{WA}}(\widehat{\mathcal{L}}_{\text{2-Loop}})}{\widehat{\pi}_{\text{WA}}(\mathcal{L}_{\text{2-Loop}})} - (1 - \widehat{\pi}_{\text{WA}}(\mathcal{L}_{\text{2-Loop}}))^N - \pi_{\text{2-Loop}}(\widehat{\mathcal{L}}_{\text{2-Loop}})$$

$$\overset{(b)}{\geq} \frac{\pi_{\text{WA}}(\widehat{\mathcal{L}}_{\text{2-Loop}}) + \frac{\delta}{3n}}{\pi_{\text{WA}}(\mathcal{L}_{\text{2-Loop}}) - \frac{\delta}{3n}} - (1 - \pi_{\text{WA}}(\mathcal{L}_{\text{2-Loop}}) - \frac{\delta}{3n})^N - \pi_{\text{2-Loop}}(\widehat{\mathcal{L}}_{\text{2-Loop}})$$

$$\overset{(c)}{\geq} -\frac{2\delta}{3n \, \pi_{\text{WA}}(\mathcal{L}_{\text{2-Loop}})} - e^{-(\pi_{\text{WA}}(\mathcal{L}_{\text{2-Loop}}) + \frac{\delta}{3n})N}$$

$$\overset{(d)}{\geq} -\frac{2\delta}{3} - \frac{\delta}{3} = -\delta.$$

In the above, (a) comes from the fact that a sample from line 6 of Algorithm 1 follows the distribution $\frac{\widehat{\pi}_{\text{WA}}(\widehat{\mathcal{L}}_{\text{2-Loop}})}{\widehat{\pi}_{\text{WA}}(\mathcal{L}_{\text{2-Loop}})}$ and the failure probability of Algorithm 1 is $(1 - \widehat{\pi}_{\text{WA}}(\mathcal{L}_{\text{2-Loop}}))^N$. For (b), we use the variation distance between $\widehat{\pi}_{\text{WA}}$ and $\pi_{\text{WA}}$ due to Lemma 1 and parameters as in Theorem 1, i.e.,

$$|\widehat{\pi}_{\text{WA}}(S) - \pi_{\text{WA}}(S)| \leq \frac{\delta}{3n} \qquad \forall \, S \subseteq \mathcal{L}_{\text{2-Loop}} \cup \mathcal{L}_{\text{2-Odd}}.$$

For (c), we use $(1 - x) \leq e^{-x}$ for any $x \geq 0$ and (d) follows from Lemma 2 and $N \leq n \ln(3\delta^{-1})$. The converse $\widehat{\pi}_{\text{2-Loop}}(\widehat{\mathcal{L}}_{\text{2-Loop}}) - \pi_{\text{2-Loop}}(\widehat{\mathcal{L}}_{\text{2-Loop}}) \leq \delta$ can be done similarly by considering the complementary set $\mathcal{L}_{\text{2-Loop}} \backslash \widehat{\mathcal{L}}_{\text{2-Loop}}$. This completes the proof of Theorem 1.

## B.1 Proof of Lemma 1

First, let $P_{\text{WA}}$ denote the transition matrix of MC induced by the worm algorithm in Section 3.1. Then we are able to define the corresponding transition graph $\mathcal{G}_{\text{WA}} = (\mathcal{L}_{\text{2-Loop}} \cup \mathcal{L}_{\text{2-Odd}}, \mathcal{E}_{\text{WA}})$, where each vertex is a state of the MC, and edges are defined on state pairs with nonzero transition probability, i.e.

$$\mathcal{E}_{\text{WA}} = \{(A, A') : (A, A') \in (\mathcal{L}_{\text{2-Loop}} \cup \mathcal{L}_{\text{2-Odd}}) \times (\mathcal{L}_{\text{2-Loop}} \cup \mathcal{L}_{\text{2-Odd}}), P_{\pi_{\text{WA}}}(A, A') > 0\}.$$

Our proof makes use of the following result proved in [18].

**Theorem 4** (Schweinsberg 2002 [18])**.** *Consider an irreducible and lazy MC, with finite state space $\Omega$, transition matrix $P$ and transition graph $\mathcal{G}_P$, which is reversible with respect to the distribution $\pi$. Let $\mathcal{O} \subseteq \Omega$ be nonempty, and for each pair $(I, J) \in \Omega \times \mathcal{O}$, specify a path $\gamma_{I,J}$ in $\mathcal{G}_P$ from $I$ to $J$. Let*

$$\Gamma = \{\gamma_{I,J} : (I, J) \in \Omega \times \mathcal{O}\}$$

*denote the collection of all such paths, and let $L(\Gamma)$ be the length of longest path in $\Gamma$. For any transition $T \in \mathcal{E}_P$, let*

$$\mathcal{H}_T = \{(I, F) \in \Omega \times \mathcal{O} : T \in \gamma_{I,J}\}.$$

*Then*

$$\tau_A(\delta) \leq \left\lceil \log\left(\frac{1}{\pi(A) + \log\left(\frac{1}{\delta}\right)}\right) \right\rceil 4L(T)\Phi(\Gamma)$$

*where*

$$\Phi(\Gamma) = \max_{(A,A') \in \mathcal{E}_P} \left\{ \sum_{I, J \in \mathcal{H}_{(A,A')}} \frac{\pi(I)\pi(J)}{\pi(\mathcal{O})\pi(A)P(A, A')} \right\}.$$

To this end, we choose $\mathcal{O} = \mathcal{L}_{\text{2-Loop}}$ and we show that there exists a choice of paths $\Gamma = \{\gamma_{I,J} : (I, J) \in (\mathcal{L}_{\text{2-Loop}} \cup \mathcal{L}_{\text{2-Loop}}) \times \mathcal{L}_{\text{2-Odd}}\}$ such that

$$\Phi(\Gamma) \leq \Delta n^4, \qquad L(\Gamma) \leq m.$$

Then we obtain the statement in Lemma 1 immediately.

We begin by specifying $\Gamma$, and then proceed to the bound of $\Phi(\Gamma)$. To this end, we fix an $[n]$-valued vertex labeling of $\mathcal{G}_{\text{WA}}$. The labeling induces a lexicographical total order of the edges, which in turn induces a lexicographical total order on the set of all subgraphs of $\mathcal{G}_{\text{WA}}$. In order for the state $I \in \mathcal{L}_{\text{2-Loop}} \cup \mathcal{L}_{\text{2-Odd}}$ transit to the $J \in \mathcal{L}_{\text{2-Loop}}$, it suffices that it updates, precisely once, those edges in $I \oplus J$. In order to describe such path, we first prove that there exist a injection from $I \oplus J$ to some unique disjoint partition $I \oplus J = \cup_{i=0}^{k} C_i$, where $C_0$ is either a path or a cycle and $C_1, \cdots, C_k$ are cycles. Observe that since $J \in \mathcal{L}_{\text{2-Loop}}$, applying symmetric difference with $J$ does not change the parity of degrees of the vertices and $I \oplus J \in \mathcal{L}_{\text{2-Loop}} \cup \mathcal{L}_{\text{2-Odd}}$. First consider the case when $I \oplus J \in \mathcal{L}_{\text{2-Odd}}$. Then there exist a path between two odd-degree vertices in $I \oplus J$, since the sum of degrees over all vertices in a component is even. Among such paths, we pick $C_0$ as the path with the highest order according to the $[n]$-valued vertex labeling. Now observe that $I \oplus J \backslash C_0 \in \mathcal{L}_{\text{2-Loop}}$ is Eulerian, which can be decomposed into disjoint set of cycles. We are able to choose a $C_1, \cdots, C_k$ uniquely by recursively excluding a cycle with the highest order, i.e. we pick $C_1$ as a cycle with highest order from $I \oplus J \backslash C_0$, then pick $C_2$ from $I \oplus J \backslash C_0 \backslash C_1$ with the highes order, and so on. For the case when $I \in \mathcal{L}_{\text{2-Loop}}$, $I \oplus J \in \mathcal{L}_{\text{2-Loop}}$ is Eulerian and we can apply similar logic to obtain the unique decomposition into disjoint cycles.

Now we are ready to describe $\gamma_{I,J}$, which updates the edges in $I \oplus J$ from $C_0$ to $C_k$ in order. If $C_0$ is a path, pick an endpoint with higher order of label and update the edges in the paths by it unwinding the edges along the path until other endpoint is met. In the case of cycles, pick a vertex with highest order of label and unwind the edges by a fixed orientation. Note that during the update of cycles, the number of odd-degree vertices are at most 2, so the intermediate states are stil in $\mathcal{L}_{\text{2-Loop}} \cup L_{\text{2-Odd}}$. As a result, we have constructed a path $\gamma_{I,F}$ for each $I \in \mathcal{L}_{\text{2-Loop}} \cup \mathcal{L}_{\text{2-Odd}}$ and $J \in \mathcal{L}_{\text{2-Loop}}$ where each edge correspond to an update on $I \oplus J$ and $|\gamma_{I,F}| = |I \oplus J| \leq m$.

Next, we bound the corresponding $\Phi(\Gamma)$. First let $\mathcal{L}_{\text{4-Odd}}$ denote the set of subgraphs with exactly 4 odd-degree vertices. We define a mapping $\eta_T : \mathcal{H}_T \to \mathcal{L}_{\text{2-Loop}} \cup \mathcal{L}_{\text{2-Odd}} \cup \mathcal{L}_{\text{4-Odd}}$ by the following:

$$\eta_T(I, J) := I \oplus F \oplus (A \cup e),$$

where $T = (A, A \oplus e)$. Observe that $\eta_T(I, J)$ agrees with $I$ on the components that have already been processed, and with $J$ on the components that have not. We prove that $\eta_T$ is an injection by reconstructing $I$ and $J$ from $\eta_T(I, J)$ given $T = (A, A \oplus e)$. To this end, observe that $I \oplus F = \eta_T(I, F) \oplus (A \cup e)$ is uniquely decided from $\eta_T(I, F)$ and $(A \cup e)$. Then given $I \oplus F$, we are able to infer the decomposition $C_0, C_1, \cdots, C_k$ of $I \oplus J$ by the rules defined previously. Moreover the updated edge $e$ implies the current set $C_i$ being updated. Therefore we can infer the processed part of $I \oplus J$. Then we can recover $J$ by beginning in $A$ and unwinding the remaining edges in $I \oplus J$ that was not processed yet. Then we recover $I$ via $I = \eta_T(I, J) \oplus (A \cup e) \oplus J$ and therefore $\eta_T$ is injective.

Next, we define a metric $w_{\text{WA}}$ such that given an edge set $F$,

$$w_{\text{WA}}(F) := \prod_{e \in F} |w(e)|.$$

We complete the proof by showing that for any $T = (A, A') \in \mathcal{E}$, the following inequality holds:

$$\Phi(\Gamma) \overset{(a)}{\leq} \sum_{I,J \in \mathcal{H}_T} \frac{1}{\pi(\mathcal{L}_{\text{2-Loop}})} \frac{\pi(I)\pi(J)}{\pi(A)P(A, A')} \overset{(b)}{\leq} \sum_{I,J \in \mathcal{H}_T} \frac{2\Delta}{w_{\text{WA}}(\mathcal{L}_{\text{2-Loop}})} \Psi(I) w_{\text{WA}}(\eta_T(I, J)) \overset{(c)}{\leq} \Delta n^4.$$

First, (a) holds by definition of $\Phi$. We prove (b) by the following chain of inequality:

$$\frac{1}{\pi(\mathcal{L}_{\text{2-Loop}})} \frac{\pi(I)\pi(J)}{\pi(A)P(A, A')} = \frac{1}{n w_{\text{WA}}(\mathcal{L}_{\text{2-Loop}})} \frac{\Psi(I) w_{\text{WA}}(I) n w_{\text{WA}}(J)}{\Psi(A) w_{\text{WA}}(A) P_{\text{WA}}(A, A')}$$

$$\overset{(1)}{\leq} \frac{1}{w_{\text{WA}}(\mathcal{L}_{\text{2-Loop}})} \Psi(I) w_{\text{WA}}(I) w_{\text{WA}}(J) \frac{2\Delta}{w_{\text{WA}}(A \cup e)}$$

$$\overset{(2)}{=} \frac{2\Delta}{w_{\text{WA}}(\mathcal{L}_{\text{2-Loop}})} \Psi(I) w_{\text{WA}}(\eta_T(I, F)).$$

In the above, (1) comes from the definition of the transition probability and (2) comes from the definition of function $w_{\text{WA}}$. Finally, we prove (c). First, we have

$$\Psi(\Gamma) \leq \sum_{(I,J)\in\mathcal{H}_T} \frac{2\Delta}{w_{\text{WA}}(\mathcal{L}_{2\text{-Loop}})} \Psi(I) w_{\text{WA}}(\eta_T(I,F))$$

$$\leq \sum_{(I,J)\in\mathcal{H}_T} \frac{2\Delta}{w_{\text{WA}}(\mathcal{L}_{2\text{-Loop}})} [w_{\text{WA}}(\mathcal{L}_{2\text{-Loop}} \cup \mathcal{L}_{2\text{-Odd}}) + 2w_{\text{WA}}(\mathcal{L}_{2\text{-Loop}} \cup \mathcal{L}_{2\text{-Odd}} \cup \mathcal{L}_{4\text{-Odd}})]$$

$$= 2\Delta \left[ (n+2) + (n+2) + \frac{w_{\text{WA}}(\mathcal{L}_{2\text{-Odd}})}{w_{\text{WA}}(\mathcal{L}_{2\text{-Loop}})} + 2\frac{w_{\text{WA}}(\mathcal{L}_{4\text{-Odd}})}{w_{\text{WA}}(\mathcal{L}_{2\text{-Loop}})} \right],$$

since $\eta_T(I,J)$ is an injection on $\mathcal{L}_{2\text{-Loop}} \cup \mathcal{L}_{2\text{-Odd}} \cup \mathcal{L}_{4\text{-Odd}}$, and the set $\mathcal{L}_{2\text{-Loop}}, \mathcal{L}_{2\text{-Odd}}, \mathcal{L}_{4\text{-Odd}}$ are disjoint. Now we prove

$$\frac{w_{\text{WA}}(\mathcal{L}_{2\text{-Odd}})}{w_{\text{WA}}(\mathcal{L}_{2\text{-Loop}})} \leq \binom{n}{2} \qquad \frac{w_{\text{WA}}(\mathcal{L}_{4\text{-Odd}})}{w_{\text{WA}}(\mathcal{L}_{2\text{-Loop}})} \leq \binom{n}{4},$$

which completes the proof of Lemma 1 since $(n+2)+(n+2)+\binom{n}{2}+2\binom{n}{4} \leq \frac{n^4}{2}$. To this end, we let $\mathcal{L}_{\text{Odd}}(W)$ denote the set of generalized loops having $W$ as the set of odd degree vertices. Now observe the following inequality:

$$\sum_{F\in\mathcal{L}_{\text{Odd}}(W)} w_{\text{WA}}(F) \overset{(a)}{=} \frac{1}{2^n} \sum_{F\in\mathcal{L}} \prod_{e\in F} |w(e)| \prod_{s\in V\setminus W} (1+(-1)^{d_F(v)}) \prod_{s\in W} (1+(-1)^{d_F(v)+1})$$

$$= \frac{1}{2^n} \sum_{\sigma\in\{-1,1\}^V} \sum_{F\in\mathcal{L}} \prod_{e\in F} |w(e)| \prod_{s\in V} \sigma_v^{d_F(v)} \prod_{v\in W} \sigma_v$$

$$= \sum_{\sigma\in\{-1,+1\}^V} \prod_{e=(u,v)\in E} (1+|w(e)|\sigma_u\sigma_v) \prod_{v\in W} \sigma_v$$

$$\overset{(b)}{\geq} \sum_{\sigma\in\{-1,+1\}^V} \prod_{e=(u,v)\in E} (1+|w(e)|\sigma_u\sigma_v)$$

$$\overset{(c)}{=} \sum_{F\in\mathcal{L}_{\mathcal{L}_{2\text{-Loop}}}} w_{\text{WA}}(F).$$

In the above, (a) comes from the fact that $1+(-1)^{d_v(F)} = 2$ if $d_v(F)$ is even and 0 otherwise, so only the terms corresponding to 2-regular loop becomes non-zero. For (b), the inequality comes from the fact that $1+|w(e)|\sigma_u\sigma_v \geq 0$ and $\sigma_v \leq 1$. For (c), the equality is from the fact that $\mathcal{L}_{2\text{-Loop}} = \mathcal{L}_{\text{Odd}}(\emptyset)$. Therefore we have $\sum_{F\in L(\emptyset)} |w(F)| \geq \sum_{F\in L(W)} |w(F)|$, leading to

$$\frac{w_{\text{WA}}(\mathcal{L}_{2\text{-Odd}})}{w_{\text{WA}}(\mathcal{L}_{2\text{-Loop}})} = \frac{\sum_{W\subseteq V, |W|=2} \sum_{F\in\mathcal{L}_{\text{Odd}}(W)} |w_{\text{WA}}(F)|}{w_{\text{WA}}(\mathcal{L}_{2\text{-Loop}})} \leq \binom{n}{2},$$

and the case for $\mathcal{L}_{4\text{-Odd}}$ is done similarly. This completes the proof of Lemma 1.

## B.2 Proof of Lemma 2

Given $W \subseteq V$, we let $\mathcal{L}_{\text{Odd}}(W)$ denote the set of generalized loops having $W$ as the set of odd degree vertices. where $\text{Odd}(F)$ is the set of odd-degree vertices in $F$. Now observe the following

inequality:

$$\sum_{F\in\mathcal{L}_{\mathrm{Odd}}(W)} w_{\mathrm{WA}}(F) \overset{(a)}{=} \frac{1}{2^n} \sum_{F\in\mathcal{L}} \prod_{e\in F} |w(e)| \prod_{s\in V\setminus W}(1+(-1)^{d_F(v)}) \prod_{s\in W}(1+(-1)^{d_F(v)+1})$$

$$= \frac{1}{2^n} \sum_{\sigma\in\{-1,1\}^V} \sum_{F\in\mathcal{L}} \prod_{e\in F} |w(e)| \prod_{s\in V} \sigma_v^{d_F(v)} \prod_{v\in W} \sigma_v$$

$$= \sum_{\sigma\in\{-1,+1\}^V} \prod_{e=(u,v)\in E}(1+|w(e)|\sigma_u\sigma_v) \prod_{v\in W} \sigma_v$$

$$\overset{(b)}{\geq} \sum_{\sigma\in\{-1,+1\}^V} \prod_{e=(u,v)\in E}(1+|w(e)|\sigma_u\sigma_v)$$

$$\overset{(c)}{=} \sum_{F\in\mathcal{L}_{\mathcal{L}_{2\text{-Loop}}}} w_{\mathrm{WA}}(F).$$

In the above, (a) comes from the fact that $1 + (-1)^{d_v(F)} = 2$ if $d_v(F)$ is even and $0$ otherwise, so only the terms corresponding to 2-regular loop becomes non-zero. For (b), the inequality comes from the fact that $1 + |w(e)|\sigma_u\sigma_v \geq 0$ and $\sigma_v \leq 1$. For (c), the equality is from the fact that $\mathcal{L}_{2\text{-Loop}} = \mathcal{L}_{\mathrm{Odd}}(\emptyset)$. Therefore we have $\sum_{F\in L(\emptyset)} |w(F)| \geq \sum_{F\in L(W)} |w(F)|$, leading to

$$\frac{\sum_{F\in\mathcal{L}_{2\text{-Loop}}} \pi_{\mathrm{WA}}(F)}{\sum_{F\in\mathcal{L}_{2\text{-Loop}}\cup\mathcal{L}_{2\text{-Odd}}} \pi_{\mathrm{WA}}(F)} = \frac{n\sum_{F\in\mathcal{L}_{2\text{-Loop}}} |w_{\mathrm{WA}}(F)|}{n\sum_{F\in\mathcal{L}_{2\text{-Loop}}} |w_{\mathrm{WA}}(F)| + \sum_{W_{\mathrm{WA}}\subseteq V,|W|=2}\sum_{F\in\mathcal{L}_{\mathrm{Odd}}(W)} |w_{\mathrm{WA}}(F)|}$$

$$\geq \frac{n}{n+2\binom{n}{2}} = \frac{1}{n},$$

which completes the proof of Lemma 2.

## C   Proof of Theorem 2

First, we quantify how much samples from Algorithm 1 are necessary for estimating some non-negative real valued function $f$ on $\mathcal{L}_{2\text{-Loop}}$. To this, we state the following lemma which is a straightforward application of the known result in [8].

**Lemma 3.** *Let $f$ be a non-negative real-valued function defined on $\mathcal{L}_{2\text{-Loop}}$ and bounded above by $f_{\max} \geq 0$. Given $0 < \xi \leq 1$ and $0 < \eta \leq 1/2$, choose*

$$s \geq \frac{504\xi^{-2}\lceil\log\eta^{-1}\rceil f_{\max}}{\mathbb{E}_{\pi_{2\text{-Loop}}}[f]} \qquad N \geq 1.2n\log\frac{24f_{\max}}{\xi\mathbb{E}_{\pi_{2\text{-Loop}}}[f]},$$

$$T \geq (m-n+1)\log 2 + 4\Delta mn^4\log\frac{8f_{\max}}{\xi\mathbb{E}_{\pi_{2\text{-Loop}}}[f]},$$

*and generate 2-regular loops $F_1, F_2, \cdots F_s$ using Algorithm 1 with inputs $N$ and $T$. Then, it follows that*

$$P\left[\frac{|\frac{1}{s}\sum_i |w(F_i)| - \mathbb{E}_{\pi_{2\text{-Loop}}}(f)|}{\mathbb{E}_{\pi_{2\text{-Loop}}}(f)} \leq \xi\right] \leq 1-\eta.$$

*namely, samples of Algorithm 1 estimates $\mathbb{E}_{\pi_{2\text{-Loop}}}(f)$ within approximation ratio $1\pm\xi$ with probability at least $1 - \eta$.*

First, recall that during each stage of simulated annealing, we approximate the expectation of the function $w(F)^{1/n}$ with respect to the distribution $\pi_{2\text{-Loop}}(\beta)$, i.e.,

$$\mathbb{E}_{\pi_{2\text{-Loop}}(\beta)}\left[|w(F)|^{1/n}\right] = Z^\dagger_{2\text{-Loop}}(\beta_{i+1})/Z^\dagger_{2\text{-Loop}}(\beta_i).$$

Hence, to apply Lemma 3, we bound $\max_F |w(F)|^{1/n}$ and $\mathbb{E}_{\pi_{2\text{-Loop}}(\beta)}\left[|w(F)|^{1/n}\right]$ as follows:

$$|w(F)|^{1/n} \leq 1 \qquad \mathbb{E}_{\pi_{2\text{-Loop}}(\beta)}\left[|w(F)|^{1/n}\right] \geq w_{\min},$$

where the first inequality is due to $w(e) \leq 1$ for any $e \in E$ and the second one is from $|F| \leq n$ for any 2-regular loop $F$. Thus, from Lemma 3 with parameters

$$s \geq 18144 n^2 \varepsilon^{-2} w_{\min}^{-1} \lceil \log(6n\nu^{-1}) \rceil, \qquad\qquad N \geq 1.2n \log(144 n \varepsilon^{-1} w_{\min}^{-1}),$$
$$T \geq (m - n + 1) \log 2 + 4\Delta m n^4 \log(48 n \varepsilon^{-1} w_{\min}^{-1}),$$

on each stage, we obtain

$$P\left[\frac{|H_i - Z_{2\text{-Loop}}^\dagger(\beta_{i+1})/Z_{2\text{-Loop}}^\dagger(\beta_i)|}{Z_{2\text{-Loop}}^\dagger(\beta_{i+1})/Z_{2\text{-Loop}}^\dagger(\beta_i)} \leq \frac{\varepsilon}{6n}\right] \geq 1 - \frac{\nu}{6n}.$$

This implies that the product $\prod_i H_i$ estimates $\frac{Z_{2\text{-Loop}}^\dagger}{2^{m-n+1}}$ within approximation ratio in

$$[((1 - \varepsilon/6n)^n, (1 + \varepsilon/6n)^n] \subseteq [1 - \varepsilon/3, 1 + \varepsilon/3]$$

with probability at least $(1 - \nu/6n)^n \geq 1 - \nu/3$, i.e.,

$$P\left[\frac{|2^{m-n+1} \prod_i H_i - Z_{2\text{-Loop}}^\dagger|}{Z_{2\text{-Loop}}^\dagger} \leq \frac{\varepsilon}{3}\right] \geq 1 - \frac{\nu}{3}.$$

Next we define a non-negative real-valued random function $g$ on $\mathcal{L}_{2\text{-Loop}}$ as

$$g(F) = \begin{cases} 1 & \text{if } w(F) < 0 \\ 0 & \text{otherwise} \end{cases},$$

namely, $\mathbb{E}_{\pi_{2\text{-Loop}}}[g(F)] = P_{\pi_{2\text{-Loop}}}[w(F) < 0]$. Since $\max_F g(F) = 1$, one can apply Lemma 3 with parameters

$$s \geq 18144 \zeta(1 - 2\zeta)^{-2} \varepsilon^{-2} \lceil \log(3\nu^{-1}) \rceil, \qquad\qquad N \geq 1.2n \log(144 \varepsilon^{-1}(1 - 2\zeta)^{-1}),$$
$$T \geq (m - n + 1) \log 2 + 4\Delta m n^4 \log(48 \varepsilon^{-1}(1 - 2\zeta)^{-1})$$

and have

$$P\left[\frac{|\kappa - P_{\pi_{2\text{-Loop}}}[w(F) < 0]|}{P_{\pi_{2\text{-Loop}}}[w(F) < 0]} \leq \frac{(1 - 2P_{\pi_{2\text{-Loop}}}[w(F) < 0])\varepsilon}{6 P_{\pi_{2\text{-Loop}}}[w(F) < 0]}\right] \geq 1 - \frac{\nu}{3},$$

since $\zeta = P_{\pi_{2\text{-Loop}}}[w(F) < 0]$. Furthermore, after some algebraic calculations, one can obtain

$$P\left[\frac{|(1 - 2\kappa) - (1 - 2P_{\pi_{2\text{-Loop}}}[w(F) < 0])|}{1 - 2P_{\pi_{2\text{-Loop}}}[w(F) < 0]} \leq \frac{\varepsilon}{3}\right] \geq 1 - \frac{\nu}{3}.$$

The rest of the proof is straightforward since we estimate $Z_{2\text{-Loop}} = (1 - 2P_{\pi_{2\text{-Loop}}}[w(F) < 0])Z_{2\text{-Loop}}^\dagger$ by $(1 - 2\kappa)2^{m-n+1} \prod_i H_i$, the approximation ratio is in $[(1 - \varepsilon/3)^2, (1 + \varepsilon/3)^2] \subseteq [1 - \varepsilon, 1 + \varepsilon]$ with probability at least $(1 - \nu/3)^2 \geq 1 - \nu$.

## D    Proof of Theorem 3

Given $F \in \mathcal{L}$, we let the odd-degree vertices in $F$ (i.e., $d_F(\cdot)$ is odd) by $v_1, v_2, \cdots v_{2\ell}$ for some integer $\ell \geq 0$. Since we assume $G$ is connected, there exist a set of paths $P_1, P_2, \cdots P_\ell$ such that $P_i$ is a path from $v_{2i-1}$ to $v_{2i}$. Note that given any set of edges $D \subseteq E$, $D \oplus P_i$ changes the parities of $d_D(v_{2i-1}), d_D(v_{2i})$, while others remain same. Therefore, all degrees in $F \oplus P_1 \oplus \cdots \oplus P_\ell$ become even. Then, due to the definition of cycle basis, there exist some $C_1, C_2, \cdots C_k \in \mathcal{C}$ such that

$$C_1 \oplus C_2 \cdots \oplus C_k = F \oplus P_1 \oplus \cdots \oplus P_\ell,$$

namely,

$$F = C_1 \oplus C_2 \cdots \oplus C_k \oplus P_1 \oplus \cdots \oplus P_\ell.$$

This completes the proof of Theorem 3.