[Reviews · NeurIPS 2016]

Reviewer 1

Summary

The paper considers approximate computation of the partition function in binary pairwise undirected graphical models. The paper introduces a novel method of combining two of the core methods, MCMC and belief propagation. The approach is based on the Loop Calculus relating the Bethe approximation to partition function (as given by BP) to the true partition function, Z_Loop= Z / Z_BP. The question thus boils down to estimating Z_Loop. Z_Loop is a huge sum over so-called generalized loops (subgraphs), and approximations can be obtained via truncation of the sum. It is shown how to sample (via MCMC, with proof of mixing/convergence time) over loops with appropriate probability (corresponding to the weight of appearance in Z_Loop) in order to approximate 2-regular Loop Sum via a simulated annealing procedure reminiscent of the multiplicative decomposition in Jerrum-Sinclair's paper. A different Markov chain is designed based on cycle basis decomposition to approximate the full (non-truncated) sum. Empirical results suggest good performance as compared to vanilla BP and MCMC approaches.

Qualitative Assessment

The problem of approximating the partition function in binary pairwise graphical models is fundamental and widely studied, yet theoretical progress has been slow outside of certain quite restricted model classes. This paper introduces a new and promising approach to the problem that appears to perform well in simulations, and may yield major theoretical progress. At the same time, certain key ingredients are proved rigorously, and the mathematical results contained in the paper are creative and nontrivial. Even the result restricted to planar graphs is interesting and insightful. It would be useful to discuss and compare to known hardness results for partition function approximation in the hard-core model and ferromagnetic Ising model. It would be nice to see simulations on (slightly) larger graphs, pushing the boundary of size for which partition function can be computed. In practice various heuristics are used for BP to improve performance (e.g. damping). Some comments on this and perhaps an empirical performance comparison would be interesting.

Confidence in this Review

3-Expert (read the paper in detail, know the area, quite certain of my opinion)


Reviewer 2

Summary

This paper presents a novel and interesting MCMC method on top of the belief propagation solution for approximating the partition function in binary graphical models. It is based on the loop calculus framework that expresses the partition function as a sum of terms, each of them corresponding to an Eulerian subgraph of the original graph. The paper explores this approach in two settings: to approximate the 2-regular partition function (a truncated series that only considers loops of degree = 2), which can be done in polynomial time for planar graphical models, but not in general and the full loop series in section 4. The algorithm combines to loop calculus with the Worm algorithm in a very nice way and derives interesting complexity results, in particular, showing that the 2-regular partition function can be efficiently approximated in non-planar graphs. The numerical evaluation perhaps is too limited, showing only results for a K_4 graph and not relating/comparing with existing literature on MC for approximation of Z. However, this is not a major limitation, given the strong algorithmic and theoretical contributions.

Qualitative Assessment

I have some minor questions: - the degree-3 constraint is not a problem as long as one has pairwise and binary variables. However, to convert a graphical model with high-order interactions to a pairwise model, usually one is required to use a non-binary alphabet. Can the authors say something about that? Is there a precise characterization of how general is the applicability of this method? - It is not clear whether the reported figures correspond to Ising models with mixed interactions. - Why not showing evaluation of Algorithm 2? For mixed interactions, sufficient large coupling strength requires to force convergence of BP. It would be interest to check the assumptions in lines 171-173 in those scenarios where standard BP does not converge. - line 159: Is the set of 2-regular loops exactly equivalent to a cycle basis? Consider two disjoint cycles: their union is a 2-regular loop, but does not belong to the cycle basis - please revise grammar LS -> "loop series" 40 : "and then run" -> "and then runs" 41 : "at the nut-shell" -> "in a nutshell" 46 : "may mix slow" -> "may mix slowly" 56 : "provide" -> "provides" 79 : "the pairwise" -> "a pairwise" 195: ")" missing ...

Confidence in this Review

3-Expert (read the paper in detail, know the area, quite certain of my opinion)


Reviewer 3

Summary

This article deals with the computation of the partition function (or normalization constant) of graphical models involving discrete variables. The approach used here is based on first, an estimation of the BP partition function (or Bethe-Peierls), which is corrected by a loop-series expansion on the considered system. The main focus of this paper is to design first, an algorithmic method computing the 2-regular generalized loops term of the expansion (which is an approximation of the true loop expansion in most cases) and converging in polynomial time. Second, they extend their algorithm in order to compute the whole loop expansion by a similar algorithm but, in this case, without providing theoretical guarantee. Finally, their second algorithm is tested on the K_4 graph (a clique of four nodes) and compared to a MCMC estimation and a BP estimation of the log of the partition function for many different types of systems : ferromagnet, antiferromagnet and disordered (random) interactions.

Qualitative Assessment

The authors develop an very interesting method to the hard problem of computing the partition function of GM. In particular, there is no method to compute Z for general graphs accurately and log(Z) is a particularly interesting component in many fields: it is related to the likelihood in Machine Learning, it gives all the statistical information of a system in statistical mechanic, ... I think therefore that any effort toward that direction is an important contribution. Concerning the work presented here, I have some questions and concerns that I would like the authors to answer: 1) the authors mentioned that BP in general fail to converge on general graph but that more involved convergent alternative exists. I think the author should be more specific, precising if all the convergent algorithms do provide the same estimation of the BP partition function and it they are indeed linear in the system size. 2) I guess that, in line 101/102, the Ising model means the 2D Ising model ? 3) I have some concern about the estimation of the full loop expansion. The 2-reg loop can be proven to converge in polynomial time but the full loop does not have these garranty. Can the author provide at least some empirical estimation of the convergence time for some systems or at least some arguments why this method will not suffer from exponential slowing down when using it on a large system ? 4) Concerning the experimental results: * it is not specified the number of systems at different temperature that were used for the estimation of the loop series. * it is not clear to me what is the MCMC estimation of log(Z), is it the estimation obtained by integrating the average energy function over \beta ? I guess not, but I think that the authors should specify how they estimate log(Z) using MCMC. * if the estimation of MCMC is not obtained by integrating over many \beta, I would like to know why it is faster to compute Z_2-loop using many different \beta instead of integrating the average energy function over many \beta ? Finally, maybe the most important point to me that would justify the whole method. The experiments are done over very small systems (4 nodes). I agree with the authors about the importance of being able to compute the true Z, but it is possible with a modest computer to reach systems with at least 20/25 nodes. Therefore I don't understand the choice of the author nor can I be sure that their method work at larger sizes. I therefore strongly advice the author to explain why such a choice. They might want as well to provide some test on larger systems comparing some estimation obtained by integration with MCMC with their method.

Confidence in this Review

2-Confident (read it all; understood it all reasonably well)


Reviewer 4

Summary

The paper is concerned with the task of estimating the partition function for a pairwise binary Markov Random Field (Ising model). The authors take the Bethe approximation resulting from Belief Propagation as a starting point and devise a simulated annealing algorithm which attempts to correct its error. This error is expressed as a loop series and the authors estimate it in two settings - one simplified where only 2-regular generalized loops are accounted for, and one full that tries to estimate the whole series. For the 2-regular case the authors base their simulated annealing on a variant of the Worm algorithm and provide theoretical convergence guarantees with high probability using a polynomial number of steps. For the general case the algorithm is based on cycle basis decomposition. The only theoretical guarantee is asymptotic convergence, but the authors show that the algorithm performs well empirically on a small Ising model.

Qualitative Assessment

The paper is technically sound as far as I can tell. Both the problem and the solution are clearly explained in technical terms. However, no wider context is provided, so not being familiar with existing literature on the topic I am unable to judge in what sort of practical settings could the proposed algorithm provide tangible improvements over existing solutions. I also find it suspicious that the paper does not discuss any other work that attempts to correct for the error introduced by the Bethe approximation - but then again I do not know any particular papers that should be used for such comparison. Some detailed comments: 1. In section 3.1, point 1., second sentence - if F is in L_2-loop then there are no odd-degree vertices in F to choose from. Is there an error in point 1? 2. Theorem 3, point 2. What happens if F+B is not in L? I assume F' <- F, but this should be clearly stated. 3. Lemma 1 should also state that abs(w(F)) = 1 iff F is empty, which is true, to support the claim that Z*_loop(infinity) = 1. Also, I believe that just above lemma 1 you mean w(0) = 1 rather than F(0) = 1. 4. The experiment is performed on a very small model (4 vertices). It would be interesting to see how the proposed algorithm performs in a reasonably large setting, ideally corresponding to a specific application.

Confidence in this Review

1-Less confident (might not have understood significant parts)


Reviewer 5

Summary

The authors propose a novel method for approximating the partition function in generalized graphical models. Improving the computation of the partition function is of great interest in Belief Propagation applications, and their sampling-based method has the advantage of allowing parameterized control over the degree of error in approximating the partition function. Their sampling approach is based on the Worm algorithm of Prokof'ev and Svistunov which is somewhat analogous to a Gibbs sampler through subgraph space in graphs with degree <= 2 (i.e. a sampler for constructing loops). The authors show that by sampling loops in this way, they can be used to estimate the Bethe approximation of the partition function. They primarily motivate their work with applications to Ising models, though they discuss applications to other general graphical models.

Qualitative Assessment

Overall, I found the potential impact of this paper very difficult to assess. This work, along with the majority of the cited prior work in this direction focus on physical models, particularly Ising models. Indeed, the experimental results for this paper are sparse, and focused solely on Ising models, though the authors claim that the method would be applicable in general to other, non-planar MRFs of arbitrary degree. Even limited in scope this way, the discussion of results in section 4.3 do little to motivate the use of this technique in general. However, even given the rather limited scope of the method, it seems that the technique borrows a number of good theoretical results from Loop Calculus and the physics community via the Worm Algorithm, and could be expected to perform quite well (as the authors do show). I think there is good potential here.

Confidence in this Review

1-Less confident (might not have understood significant parts)


Reviewer 6

Summary

I had some issues following the exposition, but my understanding is as follows: (i) We know that running BP yields marginals that can be used to compute a Bethe approximation to the partition function. However, we don't know how bad the error in this approximation can be. (ii) Correcting this estimate using the "Loop Series" is intractable, but restricting to 2-regular Loop Series is known tractable when the graphical model G is planar. (iii) The authors introduce a new polynomial-time MCMC sampling scheme for approximating Z_{2-Loop} in general pairwise binary graphical models. (iv) A similar approach is used for MCMC on general loop series to estimate the full Z_{Loop}, which is what is required to fully correct the error from the Bethe approximation.

Qualitative Assessment

+ The algorithm and analysis for MCMC for 2-regular loop series is novel and a clear improvement over the previous state of the art PTAS which only worked for planar graphs. - The paper was somewhat difficult to follow. I was not familiar with tools like Loop Calculus, and the basics of the approach were not adequately introduced or motivated. For example, why is the acronym "LS" defined on line 95? Its first mention is on line 50. - The empirical results should include approximation errors for interaction strengths with absolute value greater than 0.5. - Given the lack of a theoretical guarantee on the quality of approximation for Z_{Loop} and the missing empirical results for larger average interaction strengths it is difficult to evaluate the scenarios under which this algorithm would be useful. - There should be more diagrams explaining what the various approximations look like (in the context of graphical models).

Confidence in this Review

2-Confident (read it all; understood it all reasonably well)